

# Refining gravity anomaly data of coastal areas by combining XGM2019e-2159 and SRTM/GEBCO_2024 residual terrain model with forward modeling method

Yixiang Liu[1], Jinyun Guo[1*], Bin Guan[2], Shaofeng Bian[3], Heping Sun[4], Xin Liu[1]

5   [1]*College of Geodesy and Geomatics, Shandong University of Science and Technology, Qingdao 266590, China*;

[2]*State Key Laboratory of Spatial Datum, Xi'an 710054, China*;

[3]*Key Laboratory of Geological Survey and Evaluation of Ministry of Education, China University of Geosciences, Wuhan 430074,China*;

[4]*State Key Laboratory of Precision Geodesy,Innovation Academy for Precision Measurement Science and Technology,*

10   *Chinese Academy of Sciences, Wuhan 430077, China*

*\* Corresponding author: Jinyun Guo ( jinyunguo1@126.com)*

**Abstract.** As one of the Earth's fundamental physical fields, the gravity field model's accuracy is considerably constrained in areas with sparse coverage or data gaps. In coastal areas, satellite altimetry data are affected by land contamination and errors from tidal models, while shipborne gravity measurements fail to obtain valid gravity data in nearshore regions. Therefore, gravity field models' accuracy in coastal areas is relatively lower. Additionally, due to the truncation of global gravity field models at specific degrees, truncation errors prevent the acquisition of high-precision gravity anomaly (GA) information. In response to this problem, this study introduces detailed land topography and ocean bathymetry data, and adopts a gravity forward modeling method based on the residual terrain model (RTM) to reduce the truncation error of the gravity field model in the target coastal area. Thus, high-precision GA information can be obtained in the coastal area. First, the high-resolution terrestrial digital elevation model SRTM V4.1 is merged with the marine bathymetry model GEBCO_2024, and then combined with the reference topography model Earth2014 to construct the RTM. The RTM is then discretized into regular grid prisms, and the GA generated by the RTM at target points is computed in the spatial domain using the prism integration method to refine the XGM2019e-2159 gravity anomaly (XGM-GA) model. For computational points located in coastal areas, the rock-equivalent topography (RET) method is employed to avoid distinguishing between the different densities of land and ocean prisms during the calculation process. Based on this, a mass center offset correction is proposed to address the errors caused by prism position shifts in the RET method. To validate the feasibility of this method, this study focuses on a selected region along the U.S. West Coast (125°W–122°W, 39°N–42°N) and refines the XGM-GA model. Measured GA data from NGS99 serve as the reference for validating the experimental results. The research results show that after applying the RTM method, the root mean square error between the modeled GA and the measured GA decreased from 14.55 mGal to 8.19 mGal over the entire study area, and from 14.98 mGal to 8.19 mGal in the coastal area. The power spectral density analysis conducted at the end of this study shows that the power spectral density of the high-frequency band of the XGM-GA model



significantly increased after applying the RTM method. All the above results prove the feasibility of the RTM gravity forward modeling method in improving the accuracy of the gravity anomaly model.

## 1. Introduction

The study of the Earth's shape and its external gravity field is a primary objective of physical geodesy. The Earth's gravity field reflects the distribution and movement of mass within and on the surface of the Earth, representing a fundamental physical characteristic of our planet and serving as essential geophysical information in modern Earth sciences (Han et al., 2015; Dubey & Roy, 2023; Liang et al., 2023). High-precision gravity field models hold significant scientific and practical value in various disciplines, including geodesy, glaciology, hydrology, solid Earth geophysics, natural hazard monitoring, and resource

exploration. With the continuous development of satellite altimetry and improvements in shipborne data accuracy, the precision of marine gravity field models has been greatly enhanced (Andersen et al., 2010; Li et al., 2024; Zhou et al., 2025). However, in coastal areas, satellite altimetry data are influenced by land interference and various errors, such as those in tidal models (Hwang, 1997; Guo et al.,2010; Claessens, 2011). Furthermore, due to the distance of 5-30 km between the shipborne gravity survey lines and the coastline on the landward side, shipborne gravity measurements in this range are unable to obtain

valid data, resulting in a data gap in the coastal region (Ke et al., 2019). The widely used global gravity field models, including XGM2019e-2159 (Zingerle et al., 2020) and EGM2008 (Pavlis et al., 2012), are represented using spherical harmonic functions. These models can be used to calculate the gravity anomaly (GA) at any point on the Earth's surface and in outer space. However, due to the truncation of the spherical harmonic model at degree 2159, it cannot reflect high-frequency GA information beyond this degree(Gruber, 2009). Since the high-frequency signals of gravity field models are primarily provided

by the Earth's topography, these errors have a smaller impact in flat regions, but tend to have a larger effect in rugged mountainous areas and the coastal regions with complex terrain (Hirt, 2010). Therefore, effectively integrating topographic information into existing high-degree gravity field models is a primary method for refining regional gravity field data.

The use of detailed topography data to refine gravity field models has gained extensive research and attention in recent decades. The results of using residual terrain model (RTM) methods to calculate topographic gravity effects in rugged mountainous

areas based on high-resolution digital elevation models show that RTM methods can effectively compensate for truncation errors in GA models (Forsberg and Tscherning, 1981; Liu et al., 2025). If bathymetric data are incorporated and differences between water and crustal densities within the integration region are taken into account, the geoid model refined by RTM forward modeling with detailed topographic data can be significantly improved in accuracy (Li et al., 2024). Validation with ground-measured data showed that the high-frequency components of vertical deflections derived from RTM gravity forward

modeling can effectively compensate for the truncation errors of the EGM2008 and XGM2019e-2159 vertical deflection models (Hirt et al., 2010b; Liu et al., 2025).



Gravity forward modeling based on the RTM can be conducted in either the frequency domain (Tenzer, 2005; Yang et al., 2019; Ince et al., 2020; Wu et al., 2023) or the spatial domain (Smith, 2000; Wild-Pfeiffer, 2008; Tsoulis et al., 2009). Although the frequency-domain approach offers higher computational efficiency, its accuracy is generally lower than that of spatial domain

methods (Parker, 1995). Therefore, this study refines the gravity anomaly model for the target coastal region using the more accurate spatial-domain method. The traditional RTM method assumes a uniform density for the residual terrain within the integration region. However, if the region includes other types of landforms such as lakes, oceans, or ice sheets, this assumption of uniform prism density can lead to significant errors. In such cases, the traditional RTM method struggles to obtain a reasonable residual terrain model, necessitating improvements to meet the application requirements in complex

topographic regions. To address this, Hirt (2013) improved the traditional method by merging detailed topography and bathymetric data and adopting the rock-equivalent topography (RET) method. This approach allows for a single constant prism density within the integration region, eliminating the need to distinguish between land and ocean prisms (Kuhn and Hirt, 2016). Based on gravity forward modeling theory, topographic information can be transformed into corresponding gravity field signals. In the process of constructing the RTM from detailed and reference topography, the latter filters out the

long-wavelength components of the terrain, resulting in an RTM that retains only the high-frequency information of the topography (Hirt, 2010). The reference topographic model can be obtained either from the spherical harmonic expansion of the detailed topography or by applying a smoothing filter to the detailed model (Lin et al., 2023). When the reference topography is derived through spherical harmonic expansion of the detailed terrain using the same degree as that of the refined GA model, the GA computed from the RTM can effectively extend its high-frequency components. Generally, in areas with rugged and

complex terrain, the gravity field model lacks sufficient high-frequency GA signals, resulting in lower model accuracy.

This study primarily aims to improve the precision of the XGM2019e-2159 gravity anomaly (XGM-GA) model in coastal regions and compensate for its truncation errors. For this purpose, the RTM was first constructed using the 3″×3″ SRTM V4.1 terrestrial digital elevation data and the 15″×15″ GEBCO_2024 bathymetric data, in combination with the Earth2014 spherical harmonic reference topography model. Then, forward modeling based on the RTM is performed to obtain RTM gravity

anomalies (RTM-GA) enriched with high-frequency information, which is subsequently used to refine the XGM-GA model in the target coastal region. Finally, the NGS99 measured GA data are used as validation data to assess the effectiveness of refining the XGM-GA model using the RTM forward modeling approach.

## 2. Study area and data

### 2.1 Study area

The study area (125°W–122°W, 39°N–42°N) is located on the west coast of the United States. Since gravity forward modeling requires accounting for all topographic data within the integration region, the coverage of the topographic and bathymetric data





was extended by 1°, as shown in Fig. 1. The study area borders the Pacific Ocean to the west and includes the Central Valley, the Sierra Nevada, and the Cascade Range. The highest elevation point is Mount Shasta, located in the southern Cascade Range, with an elevation of 4316 meters. The complex topographic environment of the study area implies that the GA information

provided by global gravity field models lacks significant high-frequency GA signals.

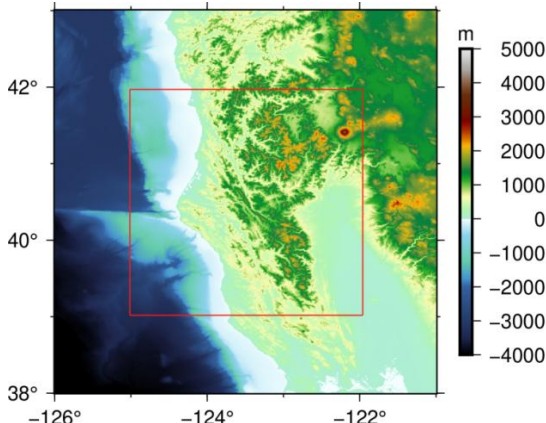

**Figure 1: Study area boundary (red box) and surrounding topography.**

### 2.2 Global Gravity Field Model

The XGM2019e-2159 global gravity field model is represented by a spherical harmonic expansion up to degree and order 2159, corresponding to a spatial resolution of $5' \times 5'$. The model is constructed based on several datasets, including the combined

satellite-only gravity field model GOCO06s, ground GA data provided by the National Geospatial-Intelligence Agency, DTU13 marine GA derived from satellite radar altimetry, and terrain gravity information over land from Earth2014. The GA model derived from XGM2019e-2159 can be computed via the International Centre for Global Earth Models (ICGEM) website (http://icgem.gfz-potsdam.de/calcgrid).

### 2.3 Digital Elevation and Bathymetric Models

The high-resolution SRTM V4.1 dataset, serving as the digital elevation model in this study, was obtained from the Shuttle Radar Topography Mission (SRTM) (https://srtm.csi.cgiar.org). This mission was a collaboration between the National Imagery and Mapping Agency and the National Aeronautics and Space Administration. The elevation data of SRTM V4.1 are referenced to the EGM96 geoid, with a spatial resolution of $3'' \times 3''$. SRTMV4.1 employs a new interpolation algorithm and supplementary DEM data to fill data voids present in SRTM3, resulting in significantly improved elevation accuracy compared

to SRTM3 (Reuter et al., 2007).

The bathymetric data in this study is sourced from GEBCO (General Bathymetric Chart of the Oceans), a project based on the Global Earth System Project. The dataset encompasses global DEM data ranging from grid scale to basin scale, integrating multiple bathymetric data sources, including shipborne echo sounding, satellite altimetry data, and other high-resolution



bathymetric measurements. The GEBCO_2024 dataset used in this study was released in July 2024 (https://www.gebco.net). It

provides globally comprehensive elevation data on a 15″×15″ geographic grid (Tozer et al., 2019).

### 2.4 Reference Topography Model

Earth2014 is a global dataset comprising topography, bathymetry, ice sheets, and high-degree spherical harmonic coefficients, developed by the Technical University of Munich and Curtin University (Hirt and Rexer, 2015). The Earth2014 dataset was constructed      using      topographic      data      from      2014      and      was      released      in      2015

(https://www.asg.ed.tum.de/iapg/forschung/topographie/earth2014). The XGM2019e-2159 model utilizes Earth2014 as its topographic data source. Earth2014 provides globally comprehensive topographic data in a 1′×1′ spatial resolution grid, making it suitable for global gravity modeling applications, particularly gravity forward modeling, geovisualization, and geophysical studies. The Earth2014 model suite is derived from four input datasets, which include elevation data for land, bedrock and ice sheets, along with bathymetric data related to lakes and oceans. The Earth2014 topographic data used in this

study are expanded in spherical harmonics up to degree 2159 in order to maintain alignment with the degree of the XGM-GA model.

### 2.5 Measured GA Data

This study uses the measured gravity data NGS99, published by the National Geodetic Survey (NGS), as the reference dataset (https://www.ncei.noaa.gov/products/gravity-data). The NGS99 dataset includes 1,633,499 discrete gravity measurement

points. The NGS99 gravity data cover not only the inland regions of the United States but also extend to its coastal areas. Figure 2 illustrates the distribution of NGS99 measured GA values and measurement locations within the study area, comprising 10,797 oceanic points and 3,247 land-based points.

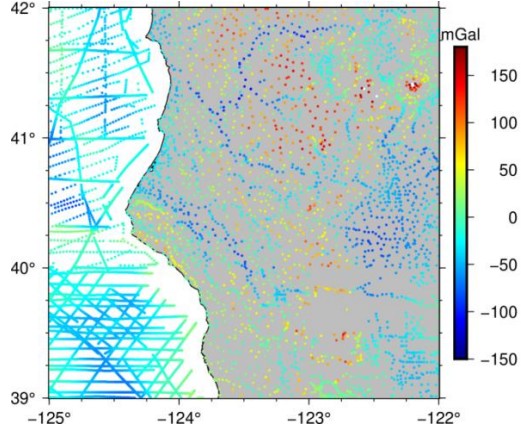

**Figure 2:** Distribution of NGS99 measured GA points in the study area.



## 3. Methodology

### 3.1 Construction of the RTM

According to the definition, RTM represents the difference between the detailed and reference topography. Due to the resolution mismatch between the 3″×3″ topographic data SRTM V4.1 on land and the 15″×15″ bathymetric data GEBCO_2024 in the ocean, and the fact that SRTM V4.1 data is only available on land, the two models must be merged before constructing the residual terrain model. First, bicubic interpolation is applied to interpolate GEBCO_2024 to a 3″×3″ grid, matching the resolution of SRTM V4.1. Then, the terrestrial data in the study area from GEBCO_2024 is removed, and the terrestrial data from SRTM V4.1 is incorporated into GEBCO_2024. This process yields the detailed topography model SRTM-GEBCO required for this study, with elevation denoted as $H^{DET}$. Thus, the RTM height is expressed as:

$$\Delta H^{\mathrm{RTM}} = H^{DET} - H^{\mathrm{REF}}, \tag{1}$$

where $H^{\mathrm{REF}}$ is expressed as follows:

$$H^{\mathrm{REF}} = \sum_{n=0}^{n_{\max}} \sum_{m=0}^{n} \left[ H_{\bar{C}_{nm}} \cos(m\lambda) + H_{\bar{S}_{nm}} \sin(m\lambda) \right] \bar{P}_{nm}(\cos\theta). \tag{2}$$

Here, $H^{REF}$ denotes the elevation of the reference topography at the computation point. The symbols $(\theta, \lambda)$ refer to the geocentric colatitude and longitude of this point. $H_{\bar{C}_{nm}}$ and $H_{\bar{S}_{nm}}$ are the fully normalized spherical harmonic coefficients describing the terrain model, with $n$ and $m$ indicating the degree and order, respectively. The function $\bar{P}_{nm}(\cos\theta)$ represents the fully normalized associated Legendre function. A schematic of the RTM is shown in Fig 3.

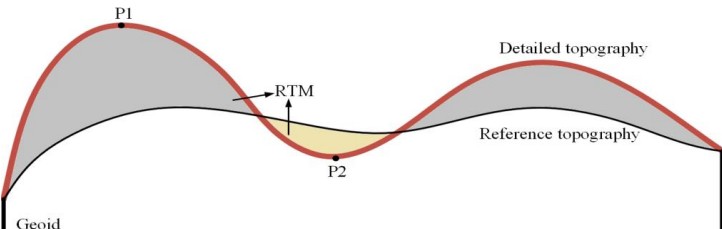

**Figure 3: Schematic diagram of the RTM.**

The residual terrain in the study area is derived using the high-resolution SRTM-GEBCO topographic model and the Earth2014 reference topography, as shown in Figure 4. A comparison between Figures 1 and 4 reveals that the residual terrain elevation exhibits alternating positive and negative values in areas with significant terrain undulation, with a maximum reaching over 800 m and a minimum below -700 m. In contrast, in oceanic regions and relatively flat plains or valleys, the variation in residual terrain elevation is also smaller.

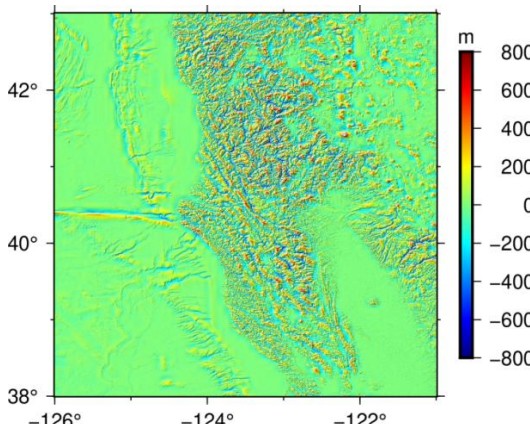

**Figure: 4 RTM of the study area.**

### 3.2 Method for Calculating RTM-GA

The GA model over coastal areas is refined in this study by applying the spatial domain methodology. The residual terrain is first segmented into discrete prism elements, and the total RTM gravity effect at each computation point is obtained by summing the contributions from all prisms within the surrounding area. Due to the oscillation of RTM elevations between negative and positive values within a certain area, gravity forward modeling based on the residual terrain model is only required over k prisms in the vicinity of the computation point (Forsberg, 1984; Hirt et al., 2010b; Wang et al., 2024). When computing derivatives of the gravitational potential such as GA and vertical deflections from the RTM, an integration radius of several tens of kilometers is generally sufficient (Hirt et al., 2010a). In this study, an integration radius of 111 km is adopted for forward modeling of the RTM-GA. To fully utilize the detailed topographic data, grid prisms with a side length of 90 m × 90 m are employed.

As shown in Fig 5, a right-handed Cartesian coordinate system is established with the Z-axis oriented vertically downward. The gravitational disturbance potential induced at point P by a prism of uniform density can be formulated as:

$$T = G\rho \int_{x_1}^{x_2} \int_{y_1}^{y_2} \int_{z_1}^{z_2} \frac{1}{R} dxdydz. \tag{3}$$

The disturbance gravity can be obtained by computing the partial derivative of the disturbance potential with respect to the vertical direction, and it is given by:

$$\delta g = -\frac{\partial T}{\partial z} = G\rho \int_{x_1}^{x_2} \int_{y_1}^{y_2} \int_{z_1}^{z_2} \frac{z-c}{R^3} dxdydz. \tag{4}$$

In this formula, G denotes the gravitational constant; $\rho$ is the density of the prism; $(x_i, y_i, z_i)$ represent the coordinates of the prism's eight corners; and $R$ is the distance from the prism's vertex to the computation point $P$, where

$$R = \sqrt{(x-a)^2 + (y-b)^2 + (z-c)^2}.$$



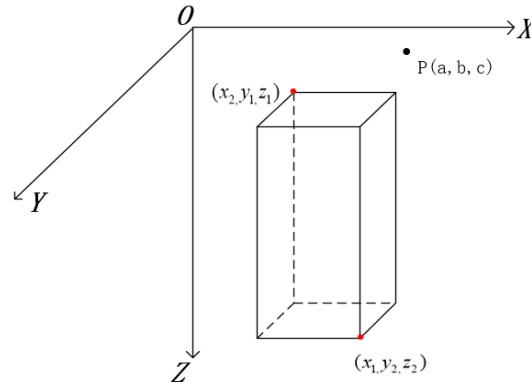

**Figure 5: Prism element model.**

After solving the integral, it can be expressed as:

$$\delta g(a,b,c) = -G\rho\{\,\|\,|\,(x-a)\ln[(y-b)+R]+(y-b)\ln[(x-a)+R]$$

$$-(z-c)\arctan\frac{(x-z)(y-b)}{(z-c)R}\,|_{x_1}^{x_2}\,|_{y_1}^{y_2}\,|_{z_1}^{z_2}\,\}\,. \tag{5}$$

The relationship between disturbance gravity and GA can be expressed as:

$$\Delta g = \delta g - \frac{2}{r}T\,. \tag{6}$$

where $r$ denotes the geocentric radius vector of the calculation point. The gravitational contribution of an individual prism to the computation point $P$ can be computed using the above formula, based on the spatial relationship between $P$ and the prism's vertices. If the integration region contains a total of $k$ prisms, the total gravitational anomaly at $P$ due to the residual

terrain is obtained by summing the contributions from all individual prisms. This yields the RTM-GA $\Delta g^{RTM}$, which is expressed as:

$$\Delta g^{RTM} = \sum_{i=1}^{k}\Delta g(i)\,. \tag{7}$$

After the RTM-GA is obtained through gravity forward modeling, the XGM-GA model can be refined, and the truncation errors can be effectively compensated. Let $\Delta g^{XGM}$ denote the modeled GA before refinement. After incorporating the $\Delta g^{XGM}$,

the refined modeled GA $\Delta g^{XGM/RTM}$ can be expressed as:

$$\Delta g^{XGM/RTM} = \Delta g^{RTM} + \Delta g^{XGM}\,. \tag{8}$$

Before performing the calculations, the geodetic coordinate system of the original topographic data needs to be transformed to the local Cartesian coordinate system centered on (123.5°W, 40.5°N) within the study area.





### 3.3 Processing of marine and coastal land areas

In inland areas, the computation points are located above the detailed topography. In this case, the density of the prism is set

to the average crustal density, $\rho_c$ = 2670 kg/m. However, over the ocean, since both the measured GA and the modeled GA

are located on the sea surface, the computation points should be placed at the sea surface rather than on the detailed seafloor

topography, as illustrated in Figure 6. Since the average seawater density is $\rho_w$ = 1030 kg/m³, the corresponding prism

density should be $\Delta\rho = \rho_c - \rho_w$.

When a terrestrial computation point is situated at the land-sea boundary, the integration region includes both land and ocean.

To avoid the need to distinguish between different density values in the forward modeling process, this study adopts the RET

method proposed by Hirt (2013). In the RET method, seawater is compressed into an equivalent rock mass by multiplying

the ocean depth ($H < 0$) with a scaling factor $(1 - \rho_w / \rho_c) \approx 0.614$. With the RET method in RTM forward modeling, both

the landmass and the compressed seawater mass can be assigned a uniform density of $\rho_c$ = 2670 kg/m³, eliminating the need

to differentiate the density of land and ocean prisms.

However, when applying the RET method to compute the RTM-GA at computation points in coastal land areas, the

compression of seawater causes a shift in the mass center of oceanic prisms, as illustrated in Fig. 6. This results in errors in

the gravity forward modeling process. Hirt (2013) suggested that in shallow coastal waters, the errors caused by this effect

are acceptable and therefore did not apply any corrections. In this study, a mass center offset correction was applied to the

oceanic prisms after RET compression. Let $H_A$ and $H_B$ denote the elevations of the detailed topography and reference

topography before compression, respectively. After applying the RET method, the elevations of the compressed detailed

topography and reference topography are denoted as $H_C$ and $H_D$, respectively. After the mass center offset correction, they

are represented as $H_Q$ and $H_W$, with their relationships expressed as follows:

$$\begin{aligned} H_C &= 0.614 H_A \\ H_D &= 0.614 H_B, \end{aligned}$$

(9)


$$\begin{aligned} H_Q &= \frac{H_A + H_B}{2} + \frac{H_C - H_D}{2} = \frac{1.614 H_A + 0.386 H_B}{2} \\ H_W &= \frac{H_A + H_B}{2} - \frac{H_C - H_D}{2} = \frac{0.386 H_A + 1.614 H_B}{2}. \end{aligned}$$

(10)



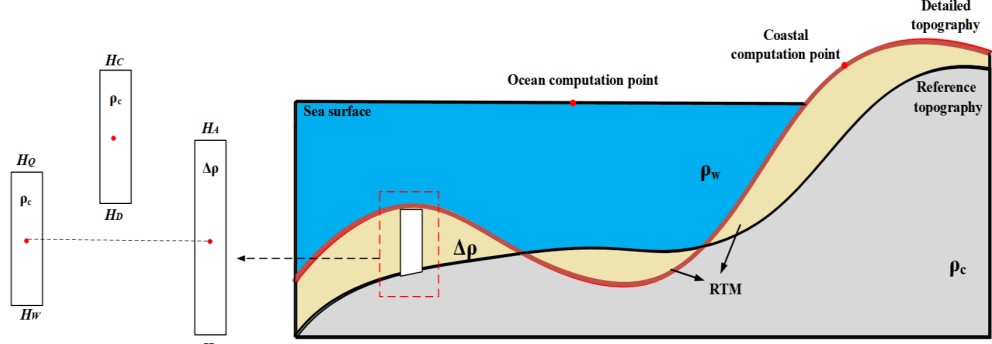

**Figure 6: Schematic diagram of oceanic computation points and mass center offset correction.**

**3.4 Harmonic Correction**

During the computation of the RTM-GA, a "non-harmonic" issue may arise when a computation point is located below the reference topography, as exemplified by point P2 in Fig. 2. For these points, the directly forward-modeled gravity potential is non-harmonic, necessitating a harmonic correction to satisfy the harmonic condition. This study employs the harmonic correction (HC) of the condensation method, which condenses the residual terrain mass between the computation point and the reference surface into an infinitely thin mass layer directly beneath the computation point. The purpose of this method is to transform this internal gravity field functional into a downward-continuous harmonic gravity field functional. This ensures that no residual mass remains above the condensed P2 point (Forsberg and Tscherning, 1981).

The harmonic correction formula of the condensation method can be expressed as:

$$HC = 4\pi G\rho_c \Delta h, \tag{11}$$

where $\rho_c$ is the average crustal density, and $\Delta h = H^{DET} - H^{REF}$ with $\Delta h < 0$. The modeled GA corrected by RTM can be expressed as:

$$\begin{cases} \Delta g^{XGM/RTM} = \Delta g^{XGM} + \Delta g^{RTM} & \Delta h >= 0 \\ \Delta g^{XGM/RTM} = \Delta g^{XGM} + \Delta g^{RTM} + HC & \Delta h < 0. \end{cases} \tag{12}$$

Since the computation points over the ocean are located on the sea surface, harmonic correction is not required in oceanic regions

Figure 7 presents the workflow for enhancing the XGM-GA model through the method described above.



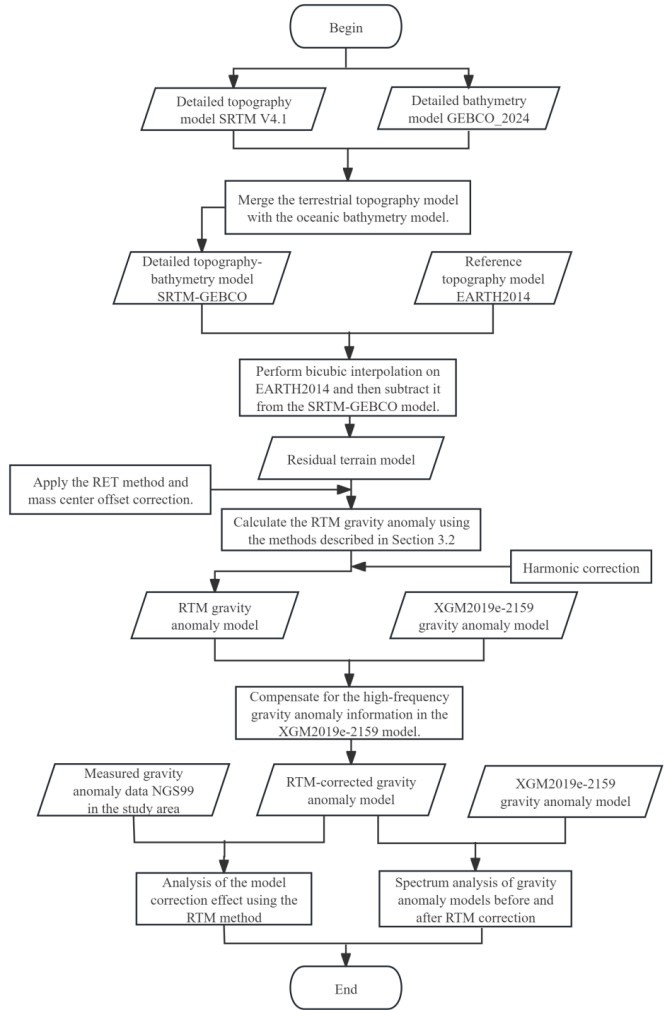

**Figure 7: Workflow of XGM-GA model refinement based on RTM.**

## 4. Experimental Results Analysis.

### 235    4.1 Computation results and overall assessment of GA

As shown in Figure 8, the 1′×1′ RTM-GA model is computed based on the residual terrain data using Equations (3)–(7).

Figure 9 shows the XGM-GA models before and after correction based on RTM gravity forward modeling. The GA model

derived from XGM2019e-2159 after RTM correction is hereafter abbreviated as the XGM/RTM-GA model.





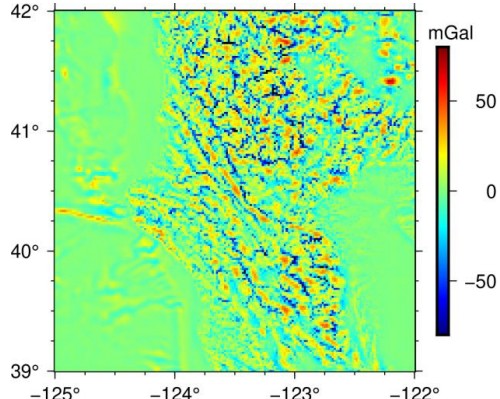

**Figure 8: RTM-GA model.**

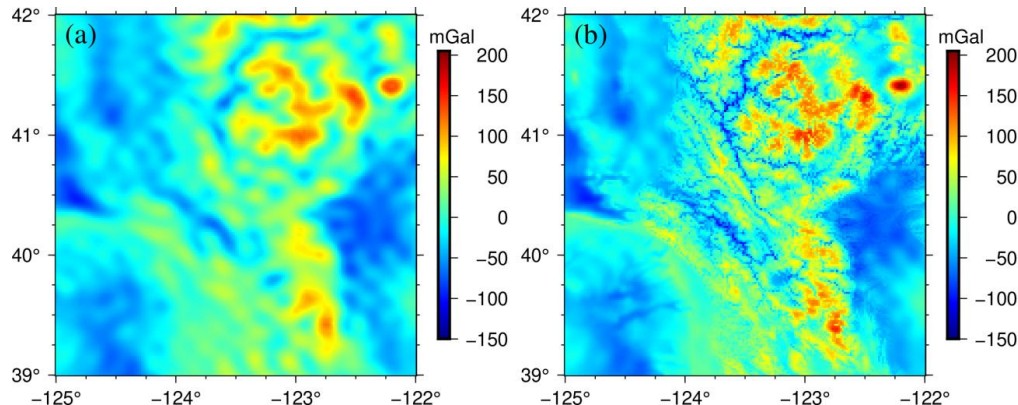

**Figure 9: XGM-GA model (a) and XGM/RTM-GA model (b).**

Table 1 presents the statistical evaluation of GA values derived from the XGM model, RTM model, and the XGM/RTM model within the study region.

**Table 1: Statistical analysis of GA models (mGal)**

| Models | Min | Max | Mean | STD | RMS |
|---|---|---|---|---|---|
| XGM | -96.4 | 138.9 | -2.96 | 39.35 | 39.46 |
| RTM | -129.65 | 73.34 | -0.80 | 17.09 | 17.11 |
| XGM/RTM | -146.29 | 204.83 | -3.76 | 43.53 | 43.69 |

Based on Figures 9 and 10 as well as Table 1, it can be concluded that both the standard deviation (STD) and the RMS of the XGM/RTM-GA model are higher than those of the XGM-GA model. The RTM-GA model primarily reflects small-scale, high-frequency GA, which are closely associated with terrain undulations. The XGM-GA model shows a smooth spatial distribution, representing large-scale gravity variations. The XGM/RTM-GA model contains both the large-scale low-frequency information of the XGM model and the small-scale high-frequency information of the RTM model, retaining the overall trend while enhancing local details.



To verify the reliability of the results, the accuracy was evaluated using the NGS99 measured GA data. Figure 10 presents the RTM-GA computed at the locations of the measured points. By comparing Figure 10 and Figure 1, it can be seen that points with large absolute values of RTM-GA are mainly located in mountainous areas with significant terrain undulations, indicating that more rugged terrain contain richer high-frequency gravity signals.

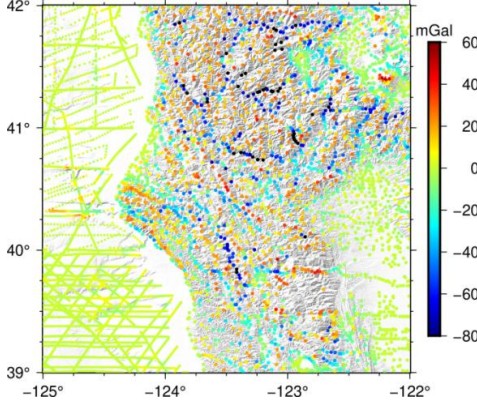

**Figure 10: RTM-GA values at the measured points.**

Figure 11(a) illustrates the computed discrepancies between the measured GA and the XGM-GA at the measured points within the study area. It is evident that the largest differences are concentrated in areas with rugged terrain within the study area, with the maximum absolute difference exceeding 100 mGal. Figure 11(b) presents the discrepancy between the measured GA and the XGM/RTM-GA.

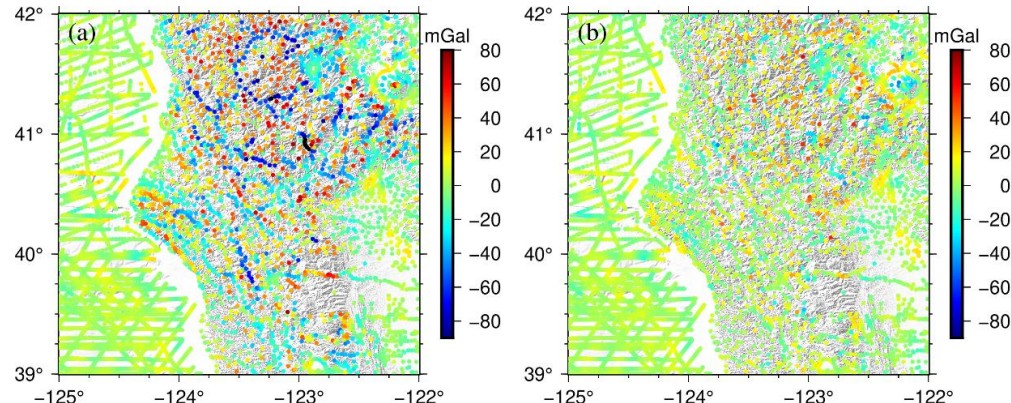

**Figure 11: The difference between the measured GA and XGM-GA (a), and the difference between the measured GA and**

**XGM/RTM-GA (b).**

A comparison between Figure 11(a) and Figure 11(b) reveals that, following RTM correction, the discrepancy between the modeled GA and the measured GA is significantly reduced in the central and northern regions with higher elevations, with a more uniform spatial distribution and a marked reduction in the number of high-amplitude areas. Incorporating the computed RTM-GA into the XGM-GA model can compensate for its truncation error, as the RTM-GA reflects high-frequency information that the XGM model cannot represent. The comparative statistical results between the XGM-GA and the




measured GA, as well as between the XGM/RTM-GA and the measured GA at land, ocean, and all measured points, are

presented in Table 2.

**Table 2. Statistical summary of discrepancies between measured and modeled GA (mGal)**

| Point type | Variant | Min | Max | Mean | STD | RMS | IR |
|---|---|---|---|---|---|---|---|
| Land point | NGS99-XGM | -111.75 | 86.77 | -4.66 | 27.67 | 28.06 | |
| | NGS99-(XGM/RTM) | -50.34 | 61.04 | 2.38 | 13.01 | 13.23 | 52.9% |
| Sea point | NGS99-XGM | -19.53 | 24.42 | 2.78 | 5.62 | 6.28 | |
| | NGS99-(XGM/RTM) | -24.26 | 20.49 | 2.37 | 5.37 | 5.87 | 6.5% |
| All point | NGS99-XGM | -111.75 | 86.77 | 1.05 | 14.58 | 14.55 | |
| | NGS99-(XGM/RTM) | -50.34 | 61.04 | 2.37 | 7.84 | 8.19 | 43.7% |

From Table 2, it can be seen that the accuracy of the GA model is improved on both land and sea after applying the RTM

correction. At the land measurement points, the RMS of the difference between the measured GA and the XGM-GA

decreased by 14.83 mGal, with an improvement rate (IR) of 52.9%. At the ocean measurement points, the RMS of the

difference decreased by 0.41 mGal, with an IR of 6.5%. Overall, at all measurement points, the RMS of the difference

decreased by 6.36 mGal, with an IR of 43.7%. According to the statistics, the accuracy of the XGM-GA model is lower in

land areas and higher in ocean regions, while the RTM correction is significantly more effective in land areas than in ocean

regions.

To investigate the correction effect of the RTM method at different elevations, the measured GA points were classified into

five categories based on elevation. The elevation ranges for these five categories are [-3400m, -1600m), [-1600m, 0m),

[0m, 500m), [500m, 1500m), and [1500m, 4000m), with 5035, 5762, 1498, 1360, and 389 points, respectively. The RTM-GA was

calculated separately for these five categories of points, and the XGM-GA and XGM/RTM-GA were compared with the

measured GA for each category. The statistical results are shown in Table 3.



**Table 3. Statistical analysis of discrepancies between measured and modeled GA across varying elevations (mGal)**

| Elevation Range | Point Numbers | Variant | Min | Max | Mean | STD | RMS |
|---|---|---|---|---|---|---|---|
| [-3400m,1600m) | 5035 | NGS99-XGM | -19.53 | 18.05 | 2.47 | 4.37 | 5.02 |
| | | NGS99(XGM/RTM) | -22.04 | 14.22 | 2.35 | 4.24 | 4.84 |
| [-1600m ,0m) | 5762 | NGS99-XGM | -18.68 | 24.42 | 3.05 | 6.51 | 7.18 |
| | | NGS99(XGM/RTM) | -24.26 | 20.49 | 2.38 | 6.19 | 6.64 |
| [0m,500m) | 1498 | NGS99-XGM | -87.52 | 29.45 | -12.63 | 20.16 | 23.79 |
| | | NGS99(XGM/RTM) | -33.62 | 48.77 | 1.57 | 9.51 | 9.64 |
| [500m,1500m) | 1360 | NGS99-XGM | -111.75 | 61.17 | -3.76 | 29.00 | 29.24 |
| | | NGS99(XGM/RTM) | -45.74 | 61.04 | 1.76 | 13.60 | 13.71 |
| [1500m,4000m) | 389 | NGS99-XGM | -75.85 | 86.77 | 22.83 | 29.93 | 37.65 |
| | | NGS99(XGM/RTM) | -50.34 | 52.96 | 7.70 | 19.74 | 21.19 |

From the table above, it can be seen that the higher the elevation on land, the greater the difference between the modeled and

measured values; the deeper the ocean, the smaller the difference at the sea surface between the modeled and measured

values. This suggests that areas with greater terrain undulation on land have lower modeled GA accuracy, as the terrain

contributes to the omission of high-frequency information in the modeled GA. At sea surface measurement points, the

shallower the water, the closer the seafloor terrain is to the measurement location. As a result, the residual seafloor

topography exerts a stronger gravitational effect at these points, leading to more effective correction results with the RTM

method. The RMS values at these five types of points were reduced by 0.18 mGal, 0.54 mGal, 14.15 mGal, 15.53 mGal, and

16.46 mGal, respectively. It is evident that as the elevation increases, the RTM correction effect also increases.

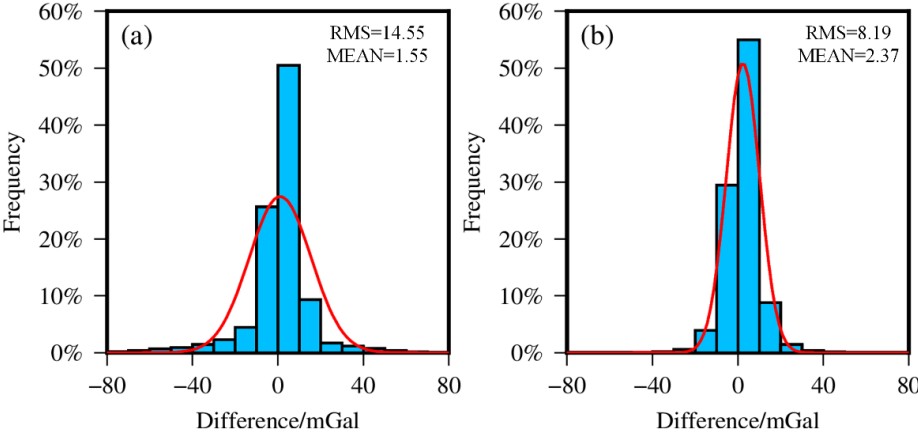

**Figure 12: Histogram of the differences between the measured GA and XGM-GA (a), and the differences between the measured**

**GA and XGM/RTM-GA (b).**

Figure 12 shows the distribution histograms of the discrepancies between the measured GA and both the XGM-GA and



XGM/RTM-GA at all measured points. Analysis results indicate that 93.75% of the deviations between the measured GA and

the XGM-GA are within ±30 mGal, whereas this percentage rises to 99.16% when compared with the XGM/RTM-GA.

These findings demonstrate a notable improvement in data quality resulting from the application of the RTM method.

**4.2 Gravity Anomaly Assessment in Coastal Land Areas**

When a terrestrial computation point is located in a coastal area, the integration region includes both land and ocean. The

standard calculation requires distinguishing rectangular prisms with different densities for land and ocean, which reduces

computational efficiency and increases the computational burden. To avoid the need to distinguish between different density

values during the forward modeling process, this study employs the RET method, compressing seawater into an equivalent

rock mass while incorporating mass center offset correction.

A 6-km landward buffer zone was established along the coastline to identify measured points where the integration region

includes both land and ocean. In the study area, 264 land-coastal points were selected for forward modeling of RTM-GA

using the RET method. The calculations considered three cases: residual terrain in the land region, residual terrain in the

ocean region, and residual terrain in the entire integration region. The statistical results are presented in Table 4.

**Table 4: GA statistics at land-coastal points (mGal)**

| Variant | RTM | Min | Max | Mean | STD | RMS | IR |
|---------|-----|-----|-----|------|-----|-----|-----|
| NGS99-XGM | Not applied | -32.62 | 49.71 | 0.75 | 14.96 | 14.98 | |
| NGS99-(XGM/RTM) | Land-only | -23.54 | 27.41 | 2.29 | 8.02 | 8.34 | 44.3% |
| NGS99-(XGM/RTM) | Sea-only | -27.71 | 49.87 | 3.02 | 13.98 | 14.30 | 4.5% |
| NGS99-(XGM/RTM) | Land/sea | -23.57 | 26.59 | 2.20 | 7.88 | 8.19 | 45.3% |

The statistical results indicate that using residual terrain on both land and ocean can refine the XGM-GA to varying degrees.

Compared to the measured GA, the RMS difference is reduced by 6.64 mGal when considering only residual terrain on land

(setting RTM elevation over the ocean to zero). When considering only residual terrain in the ocean (setting RTM elevation

over land to zero), the RMS reduction is 0.68 mGal. When considering all residual terrain, the RMS difference is reduced by

6.79 mGal.

**4.3 Power Spectral Density Analysis**

To provide a more thorough assessment of the RTM correction's influence on the XGM-GA, power spectral density (PSD)

analyses were performed in the frequency domain for both the XGM-GA and XGM/RTM-GA models. The results are shown

in Figure 13. PSD analysis is a commonly used method for evaluating signal characteristics in the frequency domain, which

effectively reveals the energy distribution of data across different frequencies. At the same wavelength, a higher PSD value

indicates that the model contains greater energy and more complete information.



According to statistical analysis, the PSD of the XGM-GA and XGM/RTM-GA models are nearly identical in the medium-to-long-wavelength range, where wavelengths exceed 39.5 km. In the wavelength range of 18.7 km to 39.5 km, the PSD of the XGM/RTM-GA model exhibits a slight increase compared to that of the XGM-GA model. For wavelengths shorter than 18.7 km, the PSD of the XGM/RTM-GA model shows a substantial enhancement relative to the XGM-GA model. These results indicate that the XGM/RTM-GA model contains more high-frequency information, confirming that the RTM method effectively compensates for the deficiency of high-frequency components in the model and reduces its truncation error.

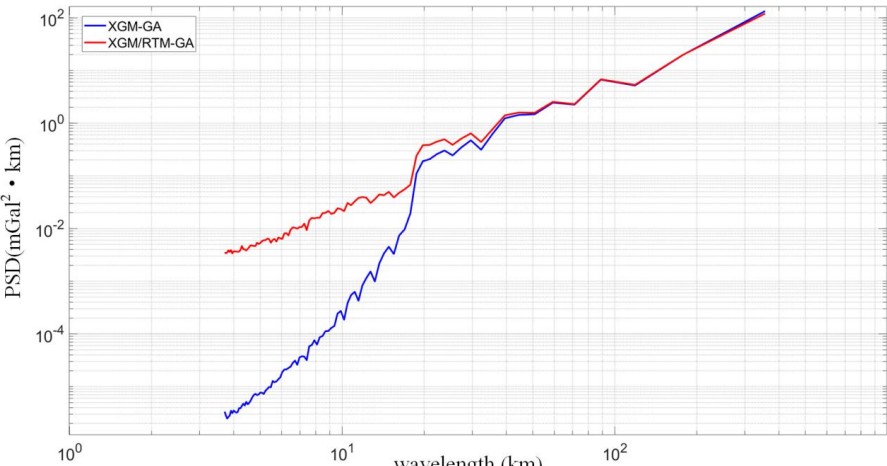

**Figure 13: PSD of the XGM-GA and XGM/RTM-GA models.**

### 5. Conclusions

In this study, the XGM-GA model in coastal areas was refined by constructing residual terrain using detailed topographic and bathymetric data. The RTM method successfully restored the high-frequency components absent in the XGM-GA model. To ensure computational efficiency and accuracy, the RET method and mass center offset correction were applied in constructing the residual terrain model for coastal areas. Consequently, the XGM/RTM-GA model was obtained for the study area. Finally, the accuracy of the XGM/RTM-GA model was validated using measured GA data from NGS99. The results show that the XGM-GA in coastal areas was significantly improved after RTM correction, with high-frequency signals effectively extended.

Statistical calculations show that after RTM correction, the XGM-GA model's precision improved by 6.36 mGal, with an IR of 43.7%. Overall, it is much more consistent with the measured values, indicating a substantial enhancement in model quality. With increasing elevation, the accuracy of the XGM-GA declines, while the magnitude of the RTM correction becomes more significant. This also confirms that terrain is an important source of high-frequency signals in GA models. At



coastal land points, the gravity forward modeling of both marine residual topography and terrestrial residual topography can

improve the accuracy of the XGM-GA to varying extents.

PSD analysis of the XGM-GA and XGM/RTM-GA models reveals a significant increase in the PSD of the GA model in the high-frequency range after RTM correction, efficiently restoring the absent high-frequency components in the XGM-GA model.

The above results confirm the effectiveness of using detailed topographic and bathymetric data to recover the

short-wavelength GA signals in gravity field models, demonstrating that the RTM forward modeling method can efficiently refine the GA information and reduce truncation errors in the model.

**Code availability**

All data used in this study are publicly available through the XGM-GA (http://icgem.gfz-potsdam.de/calcgrid), GEBCO (https://www.gebco.net), SRTM (https://srtm.csi.cgiar.org), Earth2014 (https://ddfe.curtin.edu.au/models/Earth2014), and

NGS99(https://www.ncei.noaa.gov/products/gravity-data). The source code and gravity anomaly model data before and after improvement are available at: https://doi.org/10.5281/zenodo.15300546 (Liu, 2025).

**Author contributions**

Conceptualization: YL, JG. Methodology: YL, JG. Validation: YL, JG, BG, SB, HS, XL. Writing – original draft: YL. Writing – review & editing: YL, JG, BG, SB, HS, XL. All authors contributed to writing and revising the manuscript.

**Competing interests**

The contact author has declared that none of the authors has any competing interests.

**Acknowledgements**

We express our gratitude to the following organizations: the National Aeronautics and Space Administration (NASA) and the National Imagery and Mapping Agency (NIMA) for providing the SRTM V4.1 digital elevation model, the International

Hydrographic Organization (IHO) and the Intergovernmental Oceanographic Commission of UNESCO (IOC) for providing the GEBCO-2024 bathymetric data, the National Geospatial-Intelligence Agency (NGA) for supplying the XGM2019e-2159 gravity field model, and the National Geodetic Survey (NGS) for offering the measured gravity anomaly data.

**Financial support**

This research was supported by the National Natural Science Foundation of China (grant Nos. 42430101, 42274006, and

42192535), and by the State Key Laboratory of Spatial Datum (grant No. SKLGIE2023-ZZ-5).

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
