# Peer review of "Refining gravity anomaly data of coastal areas by combining XGM2019e-2159 and SRTM/GEBCO\_2024 residual terrain model with forward modeling method"

_EGUsphere, 2025_

## Referee Comment (RC1)

In this work, the authors aimed at refining gravity anomalies of coastal areas by combining a global geopotential model (GGM) and a residual terrain model (RTM). The 3" resolution SRTM v4.1 DEM and the 15" resolution GEBCO\_2024 DBM were combined into a hybrid DEM/DBM that describes the topographic relief over the land and the seafloor undulation over the ocean. In order to avoid distinguishing the rock density and the seawater density, the rock-equivalent technique (RET) was applied to condense the seawater masses into equivalent rock masses, leading to the unified rock density within the RTM. Afterwards, the RTM gravity anomalies were estimated by the prismatic approach. The numerical results showed that improvements of gravity anomalies over both land and ocean areas could be achieved by adding RTM gravity anomalies to GGM gravity anomalies.

Refining gravity field of the GGM by incorporating high-frequency gravity field signals obtained from the RTM is widely used for modeling regional gravity field and eliminating the truncation error of the GGM, especially in the area with sparse gravity observations or even with data gaps. Obviously, the construction of an appropriate RTM is essential for this method. In the land area, since the topographic masses have the same rock density, it is straightforward to generate the RTM by subtracting a smooth reference topography from a detailed topography that is usually represented by a high-resolution DEM. However, when the research area is located nearshore, the construction of the RTM becomes more challenging as not only the topographic relief over the land should be considered but also the seafloor undulation as well as various mass densities should be taken into account. An appropriate RTM over the coastal area can also help to determine a precise land-sea (quasi)geoid model. Therefore, the motivation of this work is solid and such a work is worthy for investigation. In general, this manuscript is well organized and the English is readable. However, my primary feeling is that this work looks very similar to the work of Hirt (2013) after carefully reading through the manuscript. It seems that the novelty of this work is not obvious. Furthermore, I also have some puzzles on the procedure of constructing the RTM given in this manuscript, which are shown as follows.

**Major comments**

(1) In this manuscript, the detailed topography of the research area is described by a hybrid DEM/DBM obtained by combing the SRTM v4.1 DEM and the GEBCO\_2024 DBM, while the reference topography is derived from the Earth2014. In fact, the RTM effects represent the high-frequency part of the topographic effects, requiring that the reference topography should be

sourced from the detailed topography. Obviously, the hybrid DEM/DBM and the Earth2014 have different sources, especially in the ocean area because the GEBCO\_2024 DBM is released in 2024 but the Earth2014 is released in 2014. This might introduce additional errors due to the source inconsistency. A more proper way is to generate the reference topography by applying a spatial filtering approach (e.g. the moving average) or the spherical harmonic approach to the hybrid DEM/DBM, as what have been done in most published works of using the RTM technique. Although the numerical results are positive in this manuscript, the modeling deficiency cannot be ignored from the theoretical point of view. I cannot say the current procedure is wrong, but it is flawed.

(2) The concept of the RET is not new, and similar works have been well done in Hirt (2013). In comparison with the inland area, the ocean and coastal areas encounter more complex environments, enhancing the difficulty of constructing a proper RTM in these areas. Within this background, the main purpose of the RET is to simplify the environments by condensing seawater masses into equivalent rock masses, with the cost of changing the geometry of masses but keeping the total masses unchanged. This results in a unified topography with the same rock density over the whole research area (including the land and ocean areas). From another perspective, the application of the RET over the ocean area is actually to transform the complex oceanic environment (with both seawater and rock masses) into the simpler "land" environment (only with rock masses but with negative heights). As a result, the construction of the RTM becomes the same as the case in the land area. For example, in the work of Hirt (2013), they firstly merged the DEM and DBM into a hybrid DEM/DBM, and then applied the RET to yield a unified DEM/DBM with the same rock density. On the basis of this unified DEM/DBM, the spherical harmonic approach was applied to generate the reference topography, and finally yielding the RTM. However, I find the procedure of constructing the RTM in this manuscript is different from that of Hirt (2013). According to the workflow shown in Figure 7, it seems that the authors firstly derive an initial RTM based on the original hybrid DEM/DBM and the Earth2014 without condensing the seawater masses, and then apply the RET and mass center offset correction to the initial RTM, yielding the final RTM. I am not sure such a different procedure is due to the fact that the authors misunderstand the work of Hirt (2013) or they intend to propose a new procedure but fail to highlight it in the manuscript. If the latter reason is, the authors must highlight its novelty and compare the current results to the ones computed by the procedure of Hirt (2013). Furthermore, according to my understanding on the RTM technique, when the initial RTM is obtained without condensing the

seawater masses, its density should be the difference between the seawater density and the rock density with a positive or negative sign. Unfortunately, the authors do not give a deep and solid explanation on this issue in the manuscript. Finally, the description of the mass offset correction starting from page 9 and line 206 to Figure 6 is not clear to me. I do not understand what the authors mean and cannot follow Equations (9)-(10) as well as Figure 6. Please revise this part, at least convincing me that such a correction is reasonable and necessary.

**Minor comments**

- (1) Figure 4: Is this RTM the one obtained after applying RET? If it is not, this RTM is not the exact model that is used for computing RTM gravity anomalies. Therefore, its presentation makes no sense.
- (2) Page 7, line 170: "disturbance gravity" should be "gravity disturbance", "disturbance potential" should be "disturbing potential". Please revise similar descriptions in the other places of the manuscript.
- (3) Page 7, line 173: I do not see any  $(x_i, y_i, z_i)$  in Equations (3) and (4). Please revise it to make it more readable.
- (4) **Figure 5:**  $(x_2, y_1, z_1)$  should be  $(x_1, y_1, z_1)$ ,  $(x_1, y_2, z_2)$  should be  $(x_2, y_2, z_2)$ .
- (5) Equation (7): I suggest to revise " $\Delta g^{\text{RTM}} = \sum_{i=1}^{k} \Delta g(i)$ " to " $\Delta g^{\text{RTM}} = \sum_{i=1}^{k} \Delta g^{\text{RTM}}(i)$ ". My reason of doing so is that each prism represents the mass element of the residual terrain. So the corresponding effect is better to be marked as the RTM effect.
- (6) Page 8, line 189: If my understanding is correct, "After incorporating  $\Delta g^{\text{XGM}}$ " should be "After incorporating  $\Delta g^{\text{RTM}}$ ".

Upon the above comments, I recommend a major revision of this manuscript.

---

## Author Comment (AC1)

**Dear Reviewer:**

Thanks for your suggestions and comments. According to your suggestions, we have made major revision to the paper. Your opinions are reasonable, greatly helping me improve my article. The response to the reviewer's comments is as follows:

**Major Comments**

**Point 1:** The reviewer pointed out that the hybrid DEM/DBM and Earth2014 have different data sources, which might introduce inconsistencies, and suggested generating the reference topography from the hybrid DEM/DBM using a spatial filtering approach.

**Response 1:** We sincerely appreciate your insightful and critical comment. Indeed, the sources of the hybrid SRTM–GEBCO detailed topography/bathymetry model and the reference topography model (Earth2014) used in this study may differ, particularly between GEBCO\_2024 and Earth2014. Such source inconsistency could introduce additional errors into the modeling process.

According to your valuable suggestion, we have abandoned the use of Earth2014 as the reference topography model. Instead, a new reference topography model was generated from the SRTM–GEBCO dataset using the spatial filtering approach based on the moving average (MA) method. This modification ensures that both models share the same data source and remain consistent.

Consequently, the revised residual terrain model (RTM) was reconstructed using the newly generated reference topography. Therefore, almost all subsequent experimental results and statistical analyses have been recalculated and updated based on this improvement.

**Point 2:** The reviewer raised concerns about the procedure and steps for constructing the RTM, as well as the description of the mass center offset correction.

Response 2: (1) We sincerely appreciate your valuable comment regarding the procedure for constructing the RTM. In coastal areas, the correct RTM construction procedure should be as follows: first, the SRTM v4.1 DEM and the GEBCO\_2024 DBM are combined to generate a hybrid SRTM–GEBCO model; second, the RET method is applied to produce a unified SRTM–GEBCO dataset with a consistent rock density; third, a spatial filtering method is used to generate the reference topography; and finally, the RTM is derived. In fact, this is exactly the procedure adopted in this study. However, due to my carelessness, the workflow was incorrectly illustrated in Figure 7. I sincerely apologize for this mistake, and the figure has been corrected in the revised version.

Figure 7: Workflow of the coastal XGM-GA model refinement based on RTM

(2) Regarding the mass center offset correction, it is known that in RTM forward modeling using the RET method, both land and the condensed seawater masses can be represented using a single constant rock density. Therefore, it is unnecessary to distinguish between the density differences of land and ocean prisms.

However, in the RET process, the oceanic depth (H

Figure 5: Prism element model.

(5) Equation (7): I suggest to revise  $\Delta g^{RTM} = \sum_{i=1}^{k} \Delta g(i)$  to  $\Delta g^{RTM} = \sum_{i=1}^{k} \Delta g^{RTM}(i)$ ".

My reason of doing so is that each prism represents the mass element of the residual terrain. So the corresponding effect is better to be marked as the RTM effect.

**Response 5**: Your comment is very reasonable, and I have revised Equation (7) accordingly.

(6) Page 8, line 189: If my understanding is correct, "After incorporating  $\Delta g^{\text{XGM}}$ " should be "After incorporating  $\Delta g^{\text{RTM}}$ ".

**Response 6**: Yes, your understanding is completely correct. The error in my expression was due to my carelessness. Thank you for your comment; I have made the necessary corrections.

---

## Author Comment (AC2)

**Dear Reviewer:**

We feel great thanks for your professional review work on our paper.

As you are concerned, there are several problems that need to be addressed. According to your nice suggestions, we have made extensive corrections to our previous draft, the detailed corrections are listed below.

**(1) Accuracy of Figure References:**

The text contains inaccurate figure reference. For instance, Section 3.4 refers to "Fig. 2" in the context of illustrating computation points, but based on the context, this should likely be "Fig 3". Please conduct a thorough check to ensure all figure and table citations are correct throughout the manuscript.

**Response 1:**Thank you for your correction. In Section 3.4 of the manuscript, there was an incorrect figure reference when describing the computation points. I have made the correction and reviewed the entire manuscript. I apologize for my oversight.

**(2) Quantifying the Contribution of the Mass Center Correction:**

The proposed mass center offset correction for the RET method is a valuable improvement. However, its quantitative impact and necessity are not currently demonstrated. Providing a simple comparative result would significantly strengthen the argument for its inclusion and help readers appreciate its contribution.

Response 2: Thank you for your comment. As you pointed out, a simple

comparative analysis helps quantify the contribution of the mass center offset correction. Following your suggestion, we have added a comparative analysis in Table 4. The comparison mainly focuses on the results before and after applying the mass center offset correction(MCOC) when only the ocean RTM is considered. Relevant descriptions have also been added to the manuscript to better help readers understand the role of the mass center offset correction.

Table 4: GA statistics at land-coastal points (mGal)

| Variant         | RTM                     | Min    | Max   | Mean | STD   | RMS   | IR    |
|-----------------|-------------------------|--------|-------|------|-------|-------|-------|
| NGS99-XGM       | Not applied             | -32.62 | 49.71 | 0.75 | 14.96 | 14.98 |       |
| NGS99-(XGM/RTM) | Land-only               | -19.51 | 30.02 | 1.98 | 8.48  | 8.71  | 41.8% |
| NGS99-(XGM/RTM) | Sea-only (without-MCOC) | -29.08 | 47.02 | 0.35 | 14.63 | 14.64 | 2.3%  |
|                 | Sea-only (with-MCOC)    | -26.54 | 49.82 | 3.01 | 13.95 | 14.27 | 4.7%  |
| NGS99-(XGM/RTM) | Land/sea                | -18.54 | 28.61 | 1.79 | 7.77  | 7.97  | 46.8% |

**(3) Justification for Using the XGM2019e Model:**

The XGM2019e model provided a version up to d/o 5399 (about 2" resolution), which theoretically contains higher-resolution signal. Please justify the choice of using only the d/o 2159 model for your modeling, or demonstrate whether the XGM/RTM-GA can achieve performance superior to that of the d/o 5399 XGM2019e model.

Response 3: Thank you for your suggestion. The reasons for using the

XGM2019e-2159 model in this study are as follows, and I have also added relevant descriptions in Section 2.2 of the manuscript.

"XGM2019e is a global gravity field model developed by integrating terrestrial gravity observations with satellite-derived data. In the high-frequency spectral range, noise contributions from both data sources must be carefully addressed. In particular, coastal regions and areas with sparse or missing ground data are more prone to noise-induced distortions in the model signal. To suppress such effects, a weighted smoothing transition strategy is applied. Nevertheless, even after these procedures, the signal-to-noise ratio (SNR) at higher harmonic degrees remains relatively low, leading to attenuation of true signals and thereby constraining the practical use of high-degree gravity field models (Zingerle et al., 2020). Consequently, adopting a limited set of spherical harmonic coefficients instead of all high-degree terms provides a better trade-off between spatial resolution and SNR. Accordingly, the XGM2019e-2159 model, expanded up to degree 2159 and corresponding to a spatial resolution of  $5' \times 5'$ , is adopted in this study."

**(4) Discussion on Computational Efficiency and Relation to Existing Models:**

The method developed in this study is targeted at the challenging coastal areas. However, for land areas, high-resolution gravity field models like

the SRTM2Gravity model by Hirt et al. (2019) already exist and could potentially reduce computational burden. While the current manuscript is complete, I would be interested in the potential for a hybrid approach in the future: leveraging existing models over land and focusing the RTM forward modeling presented here primarily on the coastal transition zone. A discussion on this prospect would be valuable.

**Response 4:**Thank you for your valuable suggestion. We have added a discussion on the prospects of integrating existing high-resolution gravity field models (e.g., SRTM2Gravity) with the method presented in this study to develop gravity field models in coastal areas. The corresponding discussion has been added to Section 5.

**(5) Validation Data Coverage and Diversity:**

I note that Fig. 2 shows a notable absence of validation points in the coast area (about 20km). Could you comment on the availability of other datasets that could potentially validate the model in this critical zone? Additionally, while the NGS99 dataset is robust, incorporating additional independent validation data (such as shipborne data) would further strengthen the reliability and generalizability of the manuscript.

**Response 5:** Thank you for your insightful comment. Along shipborne gravity survey lines near the coast, the closest distance to land ranges from 5 to 30 km, where effective shipborne gravity measurements cannot be obtained, resulting in data gaps along the coastal areas (Ke Baogui et

al., 2018). At present, retrieving ocean gravity anomalies from satellite altimetry is a common approach that can be used to validate gravity field models in coastal regions. However, due to the influence of nearshore topography and shallow-water underwater terrain, the accuracy of satellite altimetry data may still be insufficient to meet the requirements (Vignudelli et al., 2011).

For this reason, effectively integrating topographic information into existing high-degree global gravity field models is an important approach to refining gravity field information in coastal areas and obtaining high-precision gravity field models.

The nearshore measured data in the NGS99 dataset actually also come from shipborne surveys, but they are limited to the coastal areas of the United States. Moreover, the quality and quantity of shipborne data have greatly improved compared to the past. Therefore, in future refinements of gravity field models in other marine regions, the use of newly released shipborne gravity data and satellite-derived ocean gravity data can be considered. We have added the corresponding discussion on this prospect to Section 5.

"Therefore, in the future, the method presented here for computing gravity in coastal areas could be applied to the construction of high-resolution coastal gravity field models, while integrating existing high-resolution gravity field models over land. This represents a

promising direction for further research. Finally, it should be noted that the NGS99 measured data, released in 1999, are only distributed over the U.S. mainland and its adjacent coastal regions. Therefore, in future work, updated measured data should be used according to the study area, especially over marine regions, where the quality and quantity of shipborne gravity data have significantly improved. For subsequent refinements of gravity field models in oceanic areas, it is recommended to consider using newly released shipborne gravity data in combination with satellite altimetry-derived ocean gravity data."

Vignudelli, Stefano, et al., eds. Coastal altimetry. Springer Science & Business Media, 2011.